# Dual-stage framework with soft-label distillation and spatial prompting for image-text retrieval

**Ran Jin[1,2], Zhengang Li**  **[1]\*, Fang Deng[1]\*, Yanhong Zhang[1]\*, Min Luo[2]\*, Tao Jin[1], Tengda Hou[1], Chenjie Du[1], Xiaozhe Gu[1], Jie Yuan[1]**

**1** Zhejiang Wanli University, School of Big Data and Software Engineering, Ningbo, China, **2** Ningbo University of Finance & Economics, Ningbo, China

\* 2023882023@zwu.edu.cn (ZL); dfsea@zwu.edu.cn (FD); zhangyan1212@zwu.edu.cn (YZ); luomin@nbufe.edu.cn (ML)

## Abstract

Vision-language pre-training (VLP) methods have significantly advanced cross-modal tasks in recent years. However, image-text retrieval still faces two critical challenges: inter-modal matching deficiency and intra-modal fine-grained localization deficiency. These issues significantly impede the accuracy of image-text retrieval. To address these challenges, we propose a novel dual-stage training framework. In the first stage, we employ Soft Label Distillation (SLD) to align the contrastive relationships between images and texts by mitigating the overfitting problem caused by hard labels. In the second stage, we introduce Spatial Text Prompt (STP) to enhance the model's visual grounding capabilities by incorporating spatial prompt information, thereby achieving more precise fine-grained alignment. Extensive experiments on standard datasets show that our method outperforms state-of-the-art approaches in image-text retrieval.The code and supplementary files can be found at https://github.com/Leon001211/DSSLP.

## Introduction

In recent years, with the advent of the big data era, fundamental tasks in cross-modal learning—such as image captioning [1] and visual question answering [2] —have received widespread attention. Image-Text Retrieval (ITR) is a fundamental task in crossmodal learning, which aims to establish semantic associations between images and texts to perform image-to-text and text-to-image retrieval. However, current mainstream ITR methods still suffer from two major limitations: the lack of sufficient modal alignment, where images and texts are not fully aligned in the shared semantic space, resulting in suboptimal cross-modal feature fusion; and the insufficient fine-grained localization capability, where models struggle to accurately identify key image regions corresponding to relevant textual content, particularly in complex scenes with multiple semantic entities. Although these problems can be described independently,

**Data availability statement:** All data and materials supporting the findings of this study are openly available in the following GitHub repository:https://github.com/Leon001211/DSSLP.

**Funding:** This work was supported by the General Scientific Research Project of Zhejiang Provincial Department of Education (Grant No. Y202456677) to Z.L., with a financial allocation of CNY 5,000. The funder provided financial support for the study design.

**Competing interests:** The authors have declared that no competing interests exist.

they are inherently interrelated—poor modal alignment exacerbates localization difficulties, while inaccurate localization further hampers effective semantic alignment.

Vision -language pre-training (VLP) models such as CLIP [3], BLIP [4] and ALIGN [5] have demonstrated strong performance by first being trained on large-scale datasets and then fine-tuned for specific downstream tasks. As a result, most existing ITR models are built upon these VLP frameworks.

In these model approaches, image-text contrastive learning is typically employed. The goal is to bring image-text pairs with semantically close relationships closer together, while pushing those that are mismatched or unrelated farther apart. However, traditional contrastive learning often misclassifies semantically relevant image-text pairs as mismatches, thereby pushing them apart in the embedding space. As illustrated in Fig 1, three image-text pairs are exemplified, which are semantically aligned, yet are treated as mismatches during the training phase, indicating a deficiency in modal alignment. This issue occurs because, in conventional contrastive learning, a pair such as (Fig 1, Caption1) is considered a positive sample, while all other captions (Caption2, Caption3) are regarded as negatives with respect to Fig 1—even though some of them are semantically matching. To address this issue, our proposed two-stage training framework employs a soft label approach, allowing for more nuanced supervision. Furthermore, we incorporate a curriculum [6] learning strategy to progressively refine the training process, enabling the model to learn more effectively and ultimately achieve superior performance.

In VLP models, we further observe that visual grounding and localization capabilities are critical for many downstream tasks. To incorporate positional information, Anderson et al [7] proposed an attention mechanism based on Faster R-CNN, enabling the model to focus on salient objects and regions within the image. Building upon this idea, Oscar [8] leverages modern object detectors to accurately identify prominent objects in images, and utilizes the detected object tags as anchors during pre-training to enhance alignment between image regions and textual descriptions. However, these models concentrate exclusively on the specific regions within bounding boxes that are identified in images during processing, without adequately considering the potentially task-relevant information that may be contained in portions of the image outside these regions.

Textual descriptions often contain words with clear semantic grounding, yet existing methods tend to overlook the semantic role of visual regions in spatial distribution. In fact, the spatial layout of objects within an image carries rich contextual cues. However, simply introducing physical position information does not directly improve matching accuracy, as there is often a misalignment between the spatial semantics in the image and the subjective spatial understanding expressed in text. For example, the phrase "a person in front of the house" may correspond to a region that appears in the center or even the upper part of the image. To address this, we propose the use of a visual location prior map, which incorporates both regional semantics and relative spatial distribution, to assist the model in achieving context-aware region localization for better cross-modal alignment. To enable the model to accurately learn spatial information and improve localization capability, we draw inspiration from the

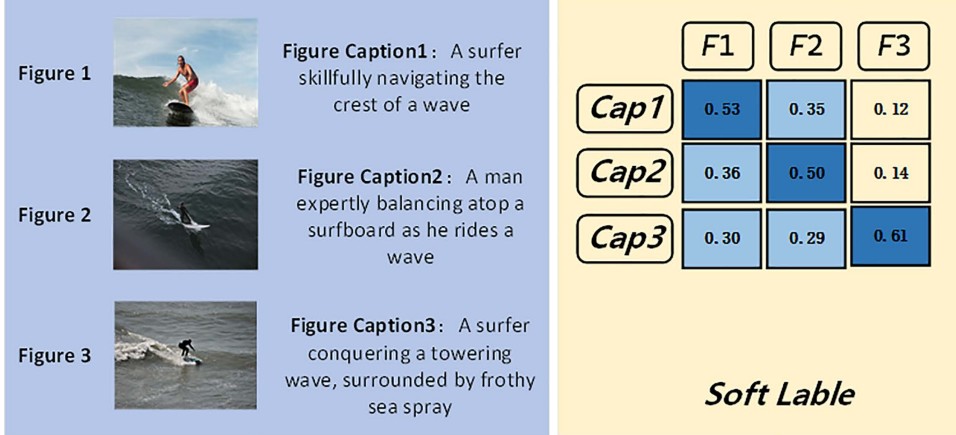

**Fig 1. Traditional contrastive learning versus soft-label contrastive learning.** Reveals the modality alignment deficiency in traditional contrastive learning, where semantically aligned image-text pairs are mistakenly treated as mismatches, causing semantically close pairs to be erroneously separated in the embedding space. Source: Unsplash, licensed under the Unsplash License.

location-aware text prompt paradigm proposed by Wang et al [9]. In our second-stage training, we integrate location-guided prompt generation into the existing task framework, and compute the loss using a soft label strategy.

Our major contributions are given below:

1. We introduce a novel alignment method named Soft Label Distillation (SLD), which utilizes soft labels as flexible supervision to facilitate cross-modal alignment in image-text retrieval (ITR) models. This approach enhances fine-grained correspondence while mitigating overfitting typically caused by rigid hard-label training.

2. We design a Spatial Text Prompt (STP) mechanism that integrates spatial cues into textual prompts, thereby boosting visual grounding precision and improving fine-grained semantic alignment between images and their corresponding text descriptions.

3. We develop a two-phase training framework and validate it through extensive evaluations on standard datasets. The findings confirm that our approach substantially enhances image-text retrieval effectiveness and achieves performance on par with state-of-the-art techniques.

## Related work

### Image-Text Retrieval

Image-Text Retrieval (ITR) is a representative cross-modal task. In recent years, a variety of models have been proposed to address this task by leveraging different techniques to achieve semantic alignment between images and texts. Early traditional methods [10–12] mainly focused on projecting features into a common subspace to maximize image-text correlation. With the rise of deep learning, more advanced models have been developed. For example, Gu et al. [13] proposed a visual-textual feature learning framework, while Ji et al. [14] introduced a Saliency-guided Attention Network (SAN) that utilizes spatial and textual attention modules to capture cross-modal correlations, thereby enhancing retrieval performance. Chen et al. [15] proposed a generalized pooling operator to extend the visual-semantic embedding model, enabling it to automatically adapt to optimal pooling strategies for different features. Many recent ITR models are built upon neural networks with attention mechanisms. For instance, Li et al. [16] proposed the VSRN (Visual Semantic Reasoning Network), which generates visual representations through regional and global semantic reasoning. In the context

of fine-grained matching, meaningless phrase-level noise is often overlooked. To address this, Diao et al. [17] proposed an effective Similarity-Aware Filtering (SAF) module to suppress irrelevant interactions. In addition, Zhang et al. [18] introduced the Negative-Aware Attention Framework, which specifically focuses on mismatched fragments to improve retrieval robustness.

Beyond these task-specific models, Vision-Language Pre-training (VLP) models have shown remarkable performance in ITR tasks. For example, CLIP [3] employs contrastive learning to align image and text pairs into a shared embedding space. Oscar [8] enhances region-text alignment by detecting salient objects and using object tags as anchors during training. VisualBERT [19] learns joint representations of images and texts through self-supervised learning on large-scale datasets. These VLP models exhibit strong generalization and flexibility, making them well-suited for a wide range of vision-language tasks.

### Alignment with soft-label

Soft labels are designed to alleviate the rigid constraints imposed by hard labels and to prevent models from becoming overconfident in incorrect predictions. This technique has proven effective across a variety of tasks. For instance, in classification, Szegedy et al. [20] introduced label smoothing, which assigns small positive values to non-ground-truth classes, thereby regularizing the model. Similarly, in the field of knowledge distillation, Hinton et al. [21] proposed using the logits predicted by a teacher model as soft labels to guide the learning of a student model. These soft labels capture richer relational information among all samples, making them more informative than hard labels. Hard labels often lead to over-confident predictions, whereas the introduction of soft labels can effectively mitigate this issue and has been shown to improve performance across various tasks. Gao et al. [22] pointed out that using one-hot encoding in VLP models introduces limitations and proposed label smoothing to provide smoother gradients, emphasizing that treating all negative samples equally is inappropriate. Subsequently, Andonian et al. [23] and Gao et al. [24] adopted self-distillation frameworks in which their models served as both teacher and student, aiming to reduce the adverse effects of noisy image-text pairs. Among these, SoftCLIP specifically applied soft-label processing within the same modality. Huang et al. [25] further advanced this idea by performing alignment both within and across modalities using soft labels. Inspired by these works, our method incorporates soft-label distillation from a teacher model during both training stages, enhancing the model's performance by providing smoother supervision and better generalization.

### Prompt learning for computer vision

In CLIP [3], a hand-crafted template such as "a photo of a [CLASS]" is used to embed textual information for zero-shot prediction. However, later studies found that such fixed prompts can negatively affect accuracy on various downstream tasks. As a result, prompt learning has been widely adopted and further improved. Prompt learning strategies vary across different downstream tasks. For example, Yao et al. [26] focused on color-aware prompt learning, leveraging shared color-based referential cues in both image and text to reduce the semantic gap. Our work is more closely related to multi-modal prompt learning [27]. CoOp [28] learns a set of continuous prompt vectors by optimizing the context to enhance generalization to unseen categories and adapt to specific tasks. Building upon CoOp, several follow-up works have aimed to further improve generalization. KgCoOp (Knowledge-Guided Context Optimization) [29] and VPT (Visual Prompt Tuning) [30] generate image-conditioned prompts by incorporating both visual features and textual tokens. Later, KgCoOp [31] introduced knowledge-guided context prompts, and TCP (Textual Class Prompting) [32] leveraged Textual Knowledge Embeddings (TKE) to map high-level class knowledge into class-aware textual tokens for improved generalization. Inspired by these developments, our work introduces spatial-aware textual prompts that incorporate positional information into text prompts, aiming to enhance the model's visual grounding and localization capabilities.

 

## Proposed method

### Preliminaries

Consider a batch of N image-text pairs $\{(I_i, T_i)\}_{i=1}^{N}$ , where $(I_i, T_i)$ represents the matching relationship between each image $I_i$ and its corresponding text $T_i$. Specifically, the image $I_i$ is mapped to a normalized representation $\hat{I}_i$ through an image encoder, and the text $T_i$ is mapped to a normalized representation $\hat{T}_i$ through a text encoder.

In previous contrastive learning, InfoNCE [33] is used to align image-text pairs, aiming to enhance the similarity between positive samples (i.e., the matching image and text) while reducing the similarity between negative samples (i.e., the mismatched image and text). $c_{ij}^{i2t}$ represents the cosine similarity between $\hat{I}_i$ and $\hat{T}_j$. Thus, we can express the similarity probability between $I_i$ and $T_j$ as well as $T_i$ and $I_j$:

$$P_{ij}^{i2t} = \frac{\exp(c_{ij}^{i2t}/\tau)}{\sum_{k=1}^{N} \exp(c_{ik}^{i2t}/\tau)}$$

(1)

$$P_{ij}^{i2t} = \frac{\exp(c_{ij}^{t2i}/\tau)}{\sum_{k=1}^{N} \exp(c_{ik}^{t2i}/\tau)}$$

(2)

Among these, $\tau$ is a learnable temperature parameter. Next, we can compute a set of dispersed probability distributions $P_i^{i2t} = (P_{i1}^{i2t}, ..., P_{iN}^{i2t})$ for each image $I_i$, and similarly, for each piece of text $T_i$, we can also obtain a set of probability distributions $P_i^{t2i} = (P_{i1}^{t2i}, ..., P_{iN}^{t2i})$. In the typical contrastive learning framework, the annotations in the dataset provide true matching information, indicating that the images and text pairs in the dataset are semantically matched, while those without annotations are assumed to be unrelated. Therefore, a one-hot encoded label vector $y_i = (y_{i1}, \dots, y_{iN})$ is introduced, where $y_{ii}$ takes the value of 1 for positive samples, and 0 in other cases. Based on this, the InfoNCE loss function can be expressed as:

$$\mathcal{L}_{itc}^{i2t} = \frac{1}{N} \sum_{i=1}^{N} \mathcal{H}(y_i, P_i^{i2t})$$

(3)

$$\mathcal{L}_{itc}^{t2i} = \frac{1}{N} \sum_{i=1}^{N} \mathcal{H}(y_i, P_i^{t2i})$$

(4)

Where, $\mathcal{H}(.,.)$ represents the cross-entropy operation, the final loss function is defined as:

$$\mathcal{L}_{itc} = \frac{\mathcal{L}_{itc}^{i2t} + \mathcal{L}_{itc}^{t2i}}{2}$$

(5)

### Cross-modal soft label distillation

We selected Unicom and Sentence-BERT as teacher models based on their strong performance in their respective domains. Unicom demonstrates robust image representation for retrieval tasks [34], while Sentence-BERT generates semantically rich text embeddings [35]. Their complementary strengths enable the generation of high-quality soft labels, which improves cross-modal alignment during distillation.

The root cause of modal alignment deficiency lies in the overly "rigid" supervisory signals used during training—strictly classifying image-text pairs as either matching or non-matching, while neglecting "suboptimal" pairs that may still exhibit potential semantic relevance. This training approach limits the model's ability to capture cross-modal semantic transfer. To address this, we introduce a soft label supervision mechanism, assigning continuous matching scores to training samples. This allows the model to differentiate between varying degrees of alignment, alleviating the semantic gap issue and enhancing cross-modal alignment capabilities.

As shown in Fig 2, We perform soft label distillation to calculate the cosine similarity between the obtained features $\hat{I}_i$ and $\hat{T}_j$, and denote it as $s_{ij}^{i2j}$. We then use batch normalization to process the probability that image $I_i$ and text $T_j$ are semantically consistent:

$$D_{ij}^{i2t} = \frac{\exp(s_{ij}^{i2t}/\tau)}{\sum\limits_{k=1}^{N} \exp(s_{ik}^{i2t}/\tau)}$$

(6)

During the training phase, $s_{ij}^{i2t}$ is used as the target distribution, and the KL divergence $P_{ij}^{i2t}$ is used to guide the learning of the distribution to achieve alignment between images and text. Similarly, $s_{ij}^{t2i}$ is also used to guide the learning of the distribution $P_{ij}^{t2i}$ to achieve alignment from text to images. Finally, the loss function for SLD is defined as:

$$\mathcal{L}_{SLD} = \frac{1}{2}(\mathcal{L}_{SLD}^{i2t} + \mathcal{L}_{SLD}^{t2i}) = \frac{1}{2}(D_{KL}(D_{ij}^{i2t} \| P_{ij}^{i2t}) + D_{KL}(D_i^{t2i} \| P_i^{t2i}))$$

(7)

Thus, the total loss function for the first phase of training is:

$$\mathcal{L}_1 = \mathcal{L}_{\text{original}} + \alpha \cdot \mathcal{L}_{SLD}$$

(8)

## Spatial text prompt

To enhance visual localization capabilities, we integrate Spatial Text Prompt (STP) into the existing VLP framework after the first-stage training, as shown in Fig 2, and conduct a second pre-training phase. This process includes two key steps:

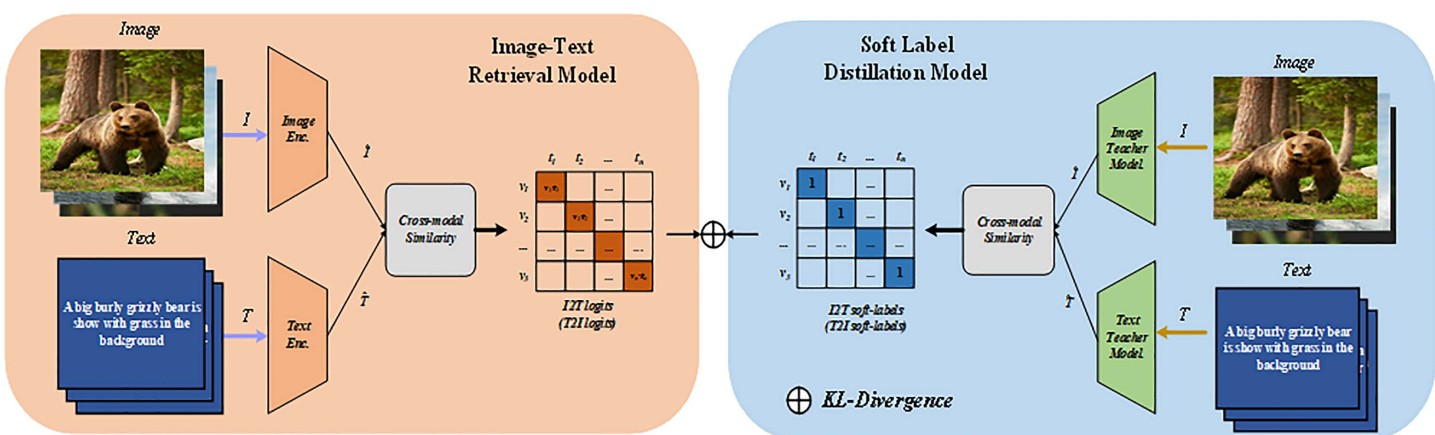

**Fig 2. Overall training framework.** Illustrates the overall architecture of our training framework, achieving image – text alignment via Soft Label Distillation. Source: Getty Images, licensed under the Unsplash+ License.

**Image**

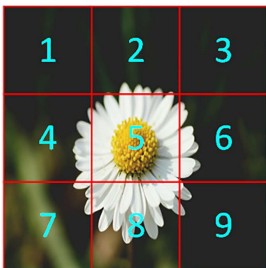

**Text**

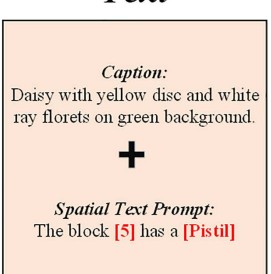

**Fig 3. Text prompt generation process.** Shows the text prompt generation process of Spatial Text Prompt (STP), which converts object labels of image blocks into a simple text format to embed location information and enhance visual localization accuracy. Source: Pexels, licensed under the Pexels License.

First, Image segmentation and object recognition, where the image is divided into multiple smaller blocks, and the objects contained within each block are identified. Second, through Text Prompt Generation, the location information of these objects is embedded into text prompts in an easily interpretable manner. By doing so, STP enables the model to learn the alignment between objects and text during the pre-training phase, allowing the model to better handle location-related visual information in subsequent downstream tasks.

**Image segmentation and object recognition.** To assign semantic labels to different spatial regions, we first extract the most frequent keywords from the training text corpus to construct a fixed vocabulary $V$. Each word in this vocabulary is then encoded into a text feature vector using CLIP's text encoder. Simultaneously, the input image is divided into $N \times N$ uniform non-overlapping blocks. For each block, we extract its visual embedding using CLIP's image encoder.

We then compute the cosine similarity between each image block feature and all keyword embeddings. The word with the highest similarity score is assigned as the pseudo-object label for that block. This approach enables efficient and scalable region-level supervision without relying on external object detectors.

**Text prompt generation.** In this phase, we transform the region-level object labels into interpretable natural language prompts. For each labeled block, a sentence is generated using a fixed template such as: "The block [S] has a [O]". where [S] denotes the spatial index of the block (e.g., a number from 1 to $N^2$, or a positional phrase like "top right"), and [O] is the corresponding object label identified in the previous step.

These spatial text prompts are concatenated with the original textual description of the image to form an enriched language input. As illustrated in Fig 3, the combined text is fed into the model's text encoder, allowing the model to learn representations that are not only semantically aligned but also spatially grounded. This procedure effectively encodes location information in a textual form, without requiring explicit coordinate embeddings or bounding boxes.

The overall pre-training process follows the framework shown in Fig 2 and enables the model to develop a more nuanced understanding of contextual and spatial cues.

## Pre-training with STP

After directly adding the Spatial Text Prompt to the original text input, our method's input label y is represented as:

$$y = [a, b] \tag{9}$$

where a denotes the original textual input and b represents the set of spatial text prompts generated through the STP mechanism. In the second stage of training, we continue to utilize the same objective as in the first stage, namely Soft Label Distillation (SLD). This method enables a more nuanced alignment between images and text by guiding the model with soft similarity scores instead of binary labels.

Specifically, for each image-text pair, we compute the cosine similarity between the image and both the original text $a$ and the spatially-augmented text $b$, and normalize these similarities using a temperature-scaled softmax. The resulting soft labels are then used to supervise the model by minimizing the KL divergence between the predicted distribution and the soft label distribution.

$$\mathcal{L}_{STP} = \frac{1}{2}(\mathcal{L}_{STP}^{i2t} + \mathcal{L}_{STP}^{t2i}) = \frac{1}{2}(D_{KL}(D_{ij}^{i2t} \parallel P_{ij}^{i2t}) + D_{KL}(D_{i}^{t2i} \parallel P_{i}^{t2i}))$$ (10)

The overall loss function for the second phase is:

$$\mathcal{L}_2 = \mathcal{L}_{\text{original}} + \beta \cdot \mathcal{L}_{\text{STP}}$$ (11)

## Experiments

### Feature extraction

To evaluate the performance of our method on image-text retrieval tasks, we conducted experimental comparisons across three widely-used datasets: Flickr30K [36], MSCOCO [37] and ECCV Caption [38] These datasets are widely used for training and performance testing of cross-modal retrieval models, with each image in the datasets accompanied by five relevant captions.

### Evaluation metrics

To assess model performance on the Flickr30K and MSCOCO datasets, we adopt the Recall@K (R@K) metric, where K is set to 1, 5, and 10. This indicator measures the fraction of queries for which the correct item appears within the top K ranked results. Since users generally prioritize the first few retrieved results, we select R@1, R@5, and R@10 as our evaluation points. These metrics effectively reflect the model's retrieval accuracy under practical search scenarios. The corresponding computation formula is provided below:

$$Recall@k = \frac{REL_k}{min(k, REL)}$$ (12)

Where $REL_k$ represents the number of relevant items in the Top-k results, and $REL$ indicates the total number of relevant items for a given query.

We also employ the R-P and mAP@R metrics to evaluate the model's ability to correctly identify and classify negative error cases on the ECCV Caption dataset. Additionally, we use the R@1 metric as a benchmark to assess the performance across different image retrieval datasets.

### Training settings

To demonstrate the effectiveness of our approach in cross-modal tasks, we performed comprehensive evaluations based on two backbone architectures: CLIP ViT-B/32 and ViT-L/14@336. The training was carried out over 30 epochs following a cosine annealing learning rate schedule, where the learning rate began at 1e-5 and gradually decreased to a minimum of 1e-6. A decay rate of 1 was applied, along with a warm-up mechanism to stabilize early-stage gradient updates. During both training phases, we incorporated the SLD and STP loss components, assigning them weights of α=0.7 and β=0.3, respectively.

## Comparison Results

To validate the effectiveness of our proposed Dual-Stage Framework with Soft-Label Distillation and Spatial Prompting (DSSLP) for image-text retrieval, we conducted extensive experiments and compared it with several state-of-the-art methods. The results demonstrate the significant performance improvements achieved by DSSLP across different datasets and backbone models.

The data for the DSSLP method presented in the tables represent the best performance metrics. As shown in Table 1 and 2, our proposed DSSLP method achieves significant performance improvements on the image-text retrieval task across both the MSCOCO and Flickr30K datasets. On the MSCOCO dataset, the DSSLP method improves the RSUM metric by 19.0% for the $CLIP_{ViT-B/32}$ model and by 8.3% for the $CLIP_{ViT-L/14@336}$ model. On the Flickr30K dataset, DSSLP similarly demonstrates excellent performance, improving the RSUM metric by 7.8% for $CLIP_{ViT-B/32}$ and by 7.6% for $CLIP_{ViT-L/14@336}$. These results highlight the effectiveness and robustness of the DSSLP strategy, as well as its broad applicability and superior performance across different models and datasets.

As shown in Table 3, on the ECCV Caption dataset, the DSSLP method also achieves significant performance improvements in the image-text retrieval task. For the two models, the average improvement in image-to-text retrieval across three metrics is 1.7%, while in text-to-image retrieval, the average improvement across all metrics is 3.0%. These results demonstrate that the introduction of the DSSLP method improves the accuracy and recall of false negative cases in the retrieval task, validating the effectiveness of our approach.

Table 4 presents the statistical robustness analysis of the DSSLP model on the MSCOCO and Flickr30K datasets. To ensure reliable evaluation, we conducted each experiment five times with different random seeds. We report the mean and standard deviation of the RSUM metric across these runs, along with p-values from paired significance tests

**Table 1. Comparison results with baselines on MSCOCO 5K.**

| Method | Image-to-Text | | | Text-to-Image | | | RSUM |
|---|---|---|---|---|---|---|---|
| | R@1 | R@5 | R@10 | R@1 | R@5 | R@10 | |
| VSE++ | 56.6 | 83.6 | 91.4 | 39.3 | 69.9 | 81.1 | 421.9 |
| NAAF | 58.9 | 85.2 | 92.0 | 42.5 | 70.9 | 81.4 | 430.9 |
| ALIGN | 58.6 | 83.0 | 89.7 | 45.6 | 69.8 | 78.6 | 425.3 |
| $CLIP_{ViT-B/32}$ | 56.3 | 81.7 | 89.4 | 42.8 | 71.2 | 81.1 | 422.6 |
| $DSSLP_{ViT-B/32}$ | 59.1 | 87.2 | 93.6 | 43.9 | 74.0 | 83.8 | 441.6 |
| $CLIP_{ViT-L/14@336}$ | 67.1 | 89.4 | 94.7 | 51.6 | 79.1 | 87.7 | 469.6 |
| $DSSLP_{ViT-L/14@336}$ | **68.2** | **90.9** | **95.0** | **53.9** | **80.8** | **89.1** | **477.9** |

**Table 2. Comparison results with baselines on Flickr30K.**

| Method | Image-to-Text | | | Text-to-Image | | | RSUM |
|---|---|---|---|---|---|---|---|
| | R@1 | R@5 | R@10 | R@1 | R@5 | R@10 | |
| VSE++ | 76.5 | 94.2 | 97.7 | 56.4 | 83.4 | 89.9 | 498.1 |
| NAAF | 81.9 | 96.1 | 98.3 | 61.0 | 85.3 | 90.6 | 513.2 |
| ALIGN | 88.6 | 98.7 | 99.7 | 75.7 | 93.8 | 96.8 | 553.3 |
| $CLIP_{ViT-B/32}$ | 78.7 | 95.4 | 98.0 | 66.3 | 88.6 | 93.1 | 520.0 |
| $DSSLP_{ViTB/32}$ | 82.1 | 95.9 | 97.8 | 68.1 | 89.7 | 94.2 | 527.8 |
| $CLIP_{ViT-L/14@336}$ | 87.3 | 99.0 | 99.5 | 76.4 | 94.8 | 97.4 | 554.5 |
| $DSSLP_{ViT-L/14@336}$ | **91.3** | **99.1** | **99.7** | **78.6** | **95.6** | **97.8** | **562.1** |

**Table 3. Experimental results of image-text retrieval on ECCV Caption.**

| Method | Image-to-Text | | | Text-to-Image | | |
|---|---|---|---|---|---|---|
| | mAP@R | R-P | R@1 | mAP@R | R-P | R@1 |
| CLIP ViT-B/32 | 28.5 | 39.4 | 72.5 | 41.7 | 50.8 | 83.0 |
| DSSLP ViT-B/32 | 29.8 | 41.2 | 73.4 | 45.1 | 53.9 | 86.0 |
| CLIP ViT-L/14@336 | 32.8 | 43.4 | 79.7 | 45.5 | 54.2 | 87.2 |
| DSSLP ViT-L/14@336 | 34.2 | 45.8 | 81.9 | 48.7 | 56.2 | 89.6 |

**Table 4. Statistical robustness analysis.**

| Dataset | Model | RSUM (Mean ± Std Dev) | p-value (vs CLIP) |
|---|---|---|---|
| MSCOCO | DSSLP ViT-B/32 | 439.8 ± 1.4 | < 0.05 |
| Flickr30K | DSSLP ViT-B/32 | 525.9 ± 0.8 | < 0.05 |

comparing DSSLP against the CLIP baseline. The results demonstrate that DSSLP consistently and significantly outperforms CLIP on both datasets, with all p-values falling below the 0.05 threshold, indicating strong statistical significance.

The ablation study results demonstrate that our proposed dual-stage training framework is effective in improving the performance of the image-text retrieval model. As shown in Table 5, we used the CLIP$_{ViT-B/32}$ model for the experiments. On both the MSCOCO and Flickr30K datasets, the addition of the second training stage significantly improves the model's retrieval accuracy. The results also highlight the importance of the dual-stage approach: the first stage allows the model to acquire foundational learning capabilities, enabling it to perform more effective learning and optimization in the second stage.

Fig 4 illustrates the normalized training loss over steps for our dual-stage framework. The x-axis represents the training steps, and the y-axis shows the normalized loss values. The graph clearly distinguishes between two training stages, marked by a vertical dashed line. In Stage 1, the loss decreases rapidly from an initial high value, indicating that the model quickly learns the inter-modal relationships through Soft Label Distillation (SLD). The loss then stabilizes, suggesting that the model has effectively captured the cross-modal features. In Stage 2, following the introduction of Spatial Text Prompting (STP), the loss experiences a further decline, which confirms the enhancement in fine-grained alignment capabilities. By the end of this stage, the loss reaches its minimum, indicating a refined understanding of complex image-text relationships by the model. The overall trend of the loss curve demonstrates the dynamic training process and highlights the effectiveness of our proposed framework in progressively improving performance across both stages.

## Conclusion

We propose a dual-stage framework to improve image-text retrieval by addressing two core challenges: cross-modal misalignment and imprecise fine-grained localization. The proposed method employs soft label distillation (SLD) and spatial text prompts (STP) to enhance the alignment between images and texts and facilitate more accurate localization of image

**Table 5. Ablation Studies on MSCOCO and Flickr30K Datasets.**

| Method | MSCOCO RSUM | Flickr30K RSUM |
|---|---|---|
| None | 422.6 | 520.0 |
| w/o STP | 432.3 | 522.3 |
| w/o SLD | 436.7 | 525.2 |
| ALL | 441.6 | 527.8 |

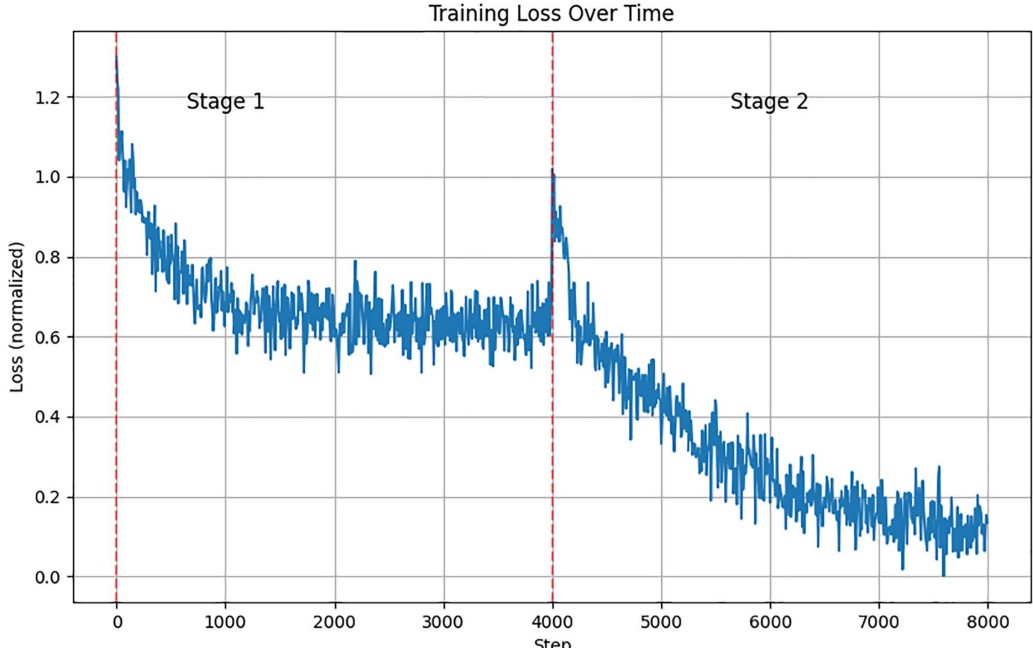

**Fig 4. Normalized Training Loss Variation in the Dual-Stage Frame-work.** Illustrates the normalized training loss over steps for our dual-stage framework.

content. The experimental outcomes show that our approach yields substantial performance gains compared to existing methods. Our method not only attains state-of-the-art results in the image-text retrieval task but also improves the model's fine-grained comprehension of image content, laying a strong foundation for future research and applications in cross-modal tasks.

## Author contributions

**Conceptualization:** Ran Jin.

**Funding acquisition:** Ran Jin.

**Investigation:** zhengang Li.

**Methodology:** zhengang Li.

**Project administration:** Tao Jin.

**Software:** Tengda Hou.

**Writing – original draft:** zhengang Li.

**Writing – review & editing:** Fang Deng, Yanhong Zhang, Min Luo, Chenjie Du, Xiaozhe Gu, Jie Yuan.

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
