## [Decision Letter · Decision Letter 0]

23 Jun 2025

PONE-D-25-28239Dual-Stage Framework with Soft-Label Distillation and Spatial Prompting for Image-Text RetrievalPLOS ONE

Dear Dr. Li,

Thank you for submitting your manuscript to PLOS ONE. After careful consideration, we feel that it has merit but does not fully meet PLOS ONE’s publication criteria as it currently stands. Therefore, we invite you to submit a revised version of the manuscript that addresses the points raised during the review process.

**ACADEMIC EDITOR: **

Consider reporting variance measures (e.g.: standard deviation across multiple runs) to demonstrate statistical robustness.

Provide confidence intervals or significance testing (e.g.: t-tests) to further confirm the reliability of performance differences.

Clarify whether all results are based on single-run evaluations or averaged over several seeds.

You used Unicom and Sentence-BERT as teacher models for generating soft labels, but no justification is provided for choosing these specific models over others.

We look forward to receiving your revised manuscript.

Kind regards,

Alessandro Bruno, Ph.D.

Academic Editor

PLOS ONE

Journal Requirements:

4. In the online submission form, you indicated that “Data cannot be shared publicly due to privacy concerns and confidentiality agreements associated with the study participants. However, data may be obtained from Mr. Li (contact via 447808600@qq.com) upon reasonable request and after a formal review process to ensure compliance with ethical standards and legal restrictions.”

**Additional Editor Comments:**

Dear Authors,

I recommend dealing with reviewers’ comments and remarks.

Consider reporting variance measures (e.g.: standard deviation across multiple runs) to demonstrate statistical robustness.

Providing confidence intervals or significance testing (e.g.: t-tests) would further confirm the reliability of performance differences.

Clarify whether all results are based on single-run evaluations or averaged over several seeds.

You used Unicom and Sentence-BERT as teacher models for generating soft labels, but no justification is provided for choosing these specific models over others.

Kindest regards,

A.B.

Reviewers' comments:

Reviewer's Responses to Questions

**Comments to the Author**

1. Is the manuscript technically sound, and do the data support the conclusions?

Reviewer #1: Yes

Reviewer #2: Yes

2. Has the statistical analysis been performed appropriately and rigorously? 

Reviewer #1: Yes

Reviewer #2: Yes

3. Have the authors made all data underlying the findings in their manuscript fully available?

Reviewer #1: No

Reviewer #2: Yes

4. Is the manuscript presented in an intelligible fashion and written in standard English?

Reviewer #1: Yes

Reviewer #2: Yes

5. Review Comments to the Author

Reviewer #1: Minor Revisions:

1. Data and Code Availability Transparency

While the authors state that data cannot be shared publicly due to privacy concerns, this is inconsistent with the fact that publicly available datasets such as MSCOCO and Flickr30K were used. It is essential to clearly state where the data used in this research can be accessed and to provide direct links to these standard datasets. Additionally, to ensure transparency and reproducibility, the authors should upload the implementation code, training scripts, and any pre-trained models to a public repository (e.g., GitHub) and cite it within the manuscript. If any customized data, labels, or processing pipelines were used, these should also be explained and shared when ethically and legally permissible. This correction is critical for meeting open science and reproducibility standards, especially in a journal like PLOS ONE.

2. Language and Grammatical Clarity

The manuscript, although generally understandable, contains numerous grammatical errors and awkward phrasings that affect readability and professionalism. Examples include incorrect verb forms (e.g., “is show” instead of “is shown”) and convoluted sentence constructions. A thorough revision of the manuscript’s language is required. It is highly recommended that the authors seek professional editing assistance or have the manuscript reviewed by a native English speaker.

3. Methodological Clarity on Spatial Text Prompting (STP)

The section describing the Spatial Text Prompt (STP) mechanism lacks sufficient detail and clarity. While the general idea is described, the exact process of dividing images into blocks, assigning labels to blocks, and converting these labels into spatial prompts needs to be more thoroughly explained. The authors should include a step-by-step description or pseudocode of the STP generation process and clarify how spatial positioning is encoded and fed into the model. This will help readers fully understand how STP contributes to model performance and ensure the method is reproducible by others.

4. Justification for Choice of Teacher Models

The manuscript uses Unicom and Sentence-BERT as teacher models for generating soft labels, but no justification is provided for choosing these specific models over others. The authors should briefly explain the rationale behind selecting these models highlighting their strengths, prior performance, or relevance to the task of image-text representation learning. This contextualization will strengthen the validity of the experimental setup and support the credibility of the results derived from these models.

5. Absence of Statistical Significance Analysis

Although the manuscript reports performance improvements over baseline models using metrics like R@1 and mAP@R, it does not include any statistical significance testing. This is particularly important when numerical improvements are relatively small, even if they are consistent. The authors are encouraged to conduct and report statistical significance tests (e.g., t-tests or confidence intervals) across multiple runs to ensure the reported gains are robust and not due to random variation. Including this analysis would enhance the scientific rigor of the experimental results and further validate the effectiveness of the proposed method.

Reviewer #2: The experimental design is robust. The authors evaluate their method on three widely used benchmarks—MSCOCO, Flickr30K, and ECCV Caption—using appropriate metrics. The proposed model shows consistent improvements over strong baselines (e.g., CLIP, ALIGN), confirming the utility of the dual-stage framework. The ablation study further supports the contribution of each proposed component.

However, a few enhancements could strengthen the experimental claims:

Consider reporting variance measures (e.g.: standard deviation across multiple runs) to demonstrate statistical robustness.

Providing confidence intervals or significance testing (e.g.: t-tests) would further confirm the reliability of performance differences.

Clarify whether all results are based on single-run evaluations or averaged over several seeds.

6. PLOS authors have the option to publish the peer review history of their article (what does this mean? ). If published, this will include your full peer review and any attached files.

**Do you want your identity to be public for this peer review?** For information about this choice, including consent withdrawal, please see our Privacy Policy .

Reviewer #1: **Yes: ** Shake Ibna Abir

Reviewer #2: **Yes: ** Mounica Achanta

---

## [Author Response · Author response to Decision Letter 1]

15 Jul 2025

Reviewer 1:

Comment 1: Data and Code Availability Transparency

Response:

Thank you for pointing this out. We apologize for the confusion. All datasets used in this study are publicly available. We have clarified the availability of MSCOCO and Flickr30K in the abstract and provided direct links to them. We have also uploaded the full implementation code, training scripts, and pre-trained weights to a public GitHub repository: https://github.com/Leon001211/DSSLP. This has been referenced in the manuscript to ensure transparency and reproducibility.

Comment 2: Language and Grammar Clarity

Response:

Thank you for your suggestion. We have carefully revised the manuscript to correct grammatical issues and improve clarity throughout. All inappropriate expressions and sentence structures have been rephrased to enhance readability and professionalism.

Comment 3: Methodological Clarity of Spatial Text Prompt (STP)

Response:

We have revised the section describing the STP mechanism (Section Image segmentation and object recognition and Pre-training with STP). We have provided a more detailed description of the specific process and how to input the model.

Comment 4: Rationale for Teacher Model Selection

Response:

We have added a clarification in Section Cross-modal soft label distillation to explain why we selected Unicoder and Sentence-BERT as teacher models. These models are known for their strong performance on sentence-level semantic similarity tasks, making them well-suited for generating soft labels in cross-modal alignment. Their relevance to our task is also supported by prior success in similar applications.

Comment 5: Lack of Statistical Significance Testing

Response:

We have added a new Table 4 to present statistical robustness analysis based on five independent runs. We report mean ± standard deviation of the RSUM metric and conduct paired t-tests comparing DSSLP with the CLIP baseline. All p-values are below 0.05, confirming that the improvements are statistically significant. These details are included in Section Comparison Results.

Reviewer 2:

Comment1: Report variance measures to demonstrate statistical robustness.

Response:

We now report mean ± standard deviation of performance metrics over five independent runs with different random seeds. This information is shown in the newly added Table 4.

Comment2: Provide confidence intervals or significance testing (e.g., t-test).

Response:

We have included paired t-tests to verify statistical significance of improvements over the CLIP baseline. p-values are below 0.05, as shown in Table 4.

Comment3: Clarify whether results are from a single run or averaged over multiple seeds.

Response:

We clarify that previous results reflected the best single-run performance. The revised manuscript now explicitly states this and complements it with averaged results over five different seeds.

---

## [Decision Letter · Decision Letter 1]

10 Sep 2025

Dual-Stage Framework with Soft-Label Distillation and Spatial Prompting for Image-Text Retrieval

PONE-D-25-28239R1

Dear Dr. Li,

We’re pleased to inform you that your manuscript has been judged scientifically suitable for publication and will be formally accepted for publication once it meets all outstanding technical requirements.

Kind regards,

Ming Liu

Academic Editor

PLOS ONE

Additional Editor Comments (optional):

The revisions have been addressed accordingly, recommend to accept the paper

Reviewers' comments:

Reviewer's Responses to Questions

**Comments to the Author**

1. If the authors have adequately addressed your comments raised in a previous round of review and you feel that this manuscript is now acceptable for publication, you may indicate that here to bypass the “Comments to the Author” section, enter your conflict of interest statement in the “Confidential to Editor” section, and submit your "Accept" recommendation.

Reviewer #1: All comments have been addressed

Reviewer #2: All comments have been addressed

2. Is the manuscript technically sound, and do the data support the conclusions?

Reviewer #1: Yes

Reviewer #2: Yes

3. Has the statistical analysis been performed appropriately and rigorously? 

Reviewer #1: Yes

Reviewer #2: Yes

4. Have the authors made all data underlying the findings in their manuscript fully available?

Reviewer #1: Yes

Reviewer #2: Yes

5. Is the manuscript presented in an intelligible fashion and written in standard English?

Reviewer #1: Yes

Reviewer #2: Yes

6. Review Comments to the Author

Reviewer #1: I appreciate the thoughtful and thorough revisions made to the manuscript. All my previous concerns have been carefully addressed:

Data and code availability have been clarified, with direct dataset links and an open GitHub repository now provided. The manuscript has been revised for grammar and clarity, which has improved readability.

The Spatial Text Prompt (STP) methodology is now described in detail, including block segmentation, labeling, and prompt generation, supported by clear figures. The choice of teacher models (Unicoder and Sentence-BERT) is now well justified. Statistical robustness has been added through multiple runs and paired t-tests, confirming the significance of reported improvements.

Overall, the manuscript has been substantially improved and is now suitable for publication. I recommend acceptance pending minor proofreading and typesetting.

Reviewer #2: No comments, paper is technically sound and is good to publish. The revised manuscript provides clearer explanations of STP generation, teacher model rationale, and experimental setup. Ablation studies effectively isolate the contributions of SLD and STP.

7. PLOS authors have the option to publish the peer review history of their article (what does this mean? ). If published, this will include your full peer review and any attached files.

**Do you want your identity to be public for this peer review?** For information about this choice, including consent withdrawal, please see our Privacy Policy .

Reviewer #1: **Yes: ** Shake Ibna Abir

Reviewer #2: **Yes: ** Mounica Achanta

---

## [Editor Report · Acceptance letter]

PONE-D-25-28239R1

PLOS ONE

Dear Dr. Li,

I'm pleased to inform you that your manuscript has been deemed suitable for publication in PLOS ONE. Congratulations! Your manuscript is now being handed over to our production team.

Kind regards,

on behalf of

Professor Ming Liu

Academic Editor

PLOS ONE